# Heterologous Expression of *Jatropha curcas Fatty Acyl-ACP Thioesterase A (JcFATA)* and *B (JcFATB)* Affects Fatty Acid Accumulation and Promotes Plant Growth and Development in *Arabidopsis*

**DOI:** 10.3390/ijms23084209

**Published:** 2022-04-11

**Authors:** Ying Liu, Jing Han, Zhijie Li, Zuojie Jiang, Liangfeng Luo, Yingzhe Zhang, Minghao Chen, Yuesheng Yang, Zhenlan Liu

**Affiliations:** Department of Genetics, College of Life Sciences, South China Agricultural University, Guangzhou 510642, China; liuying85168@126.com (Y.L.); jhan0227@163.com (J.H.); lizhijie5318@163.com (Z.L.); jiangzuojie5905@163.com (Z.J.); a1261594295@163.com (L.L.); jomer470569131@163.com (Y.Z.); cminghao_cmh@163.com (M.C.)

**Keywords:** fatty acid, heterologous expression, *Jatropha curcas* L., *JcFATA*, *JcFATB*

## Abstract

Plant fatty acyl-acyl carrier protein (ACP) thioesterases terminate the process of de novo fatty acid biosynthesis in plastids by hydrolyzing the acyl-ACP intermediates, and determine the chain length and levels of free fatty acids. They are of interest due to their roles in fatty acid synthesis and their potential to modify plant seed oils through biotechnology. Fatty acyl-ACP thioesterases (FAT) are divided into two families, i.e., FATA and FATB, according to their amino acid sequence and substrate specificity. The high oil content in *Jatropha curcas* L. seed has attracted global attention due to its potential for the production of biodiesel. However, the detailed effects of *JcFATA* and *JcFATB* on fatty acid biosynthesis and plant growth and development are still unclear. In this study, we found that *JcFATB* transcripts were detected in all tissues and organs examined, with especially high accumulation in the roots, leaves, flowers, and some stages of developing seeds, and *JcFATA* showed a very similar expression pattern. Subcellular localization of the JcFATA-GFP and JcFATB-GFP fusion protein in *Arabidopsis* leaf protoplasts showed that both JcFATA and JcFATB localized in chloroplasts. Heterologous expression of *JcFATA* and *JcFATB* in *Arabidopsis thaliana* individually generated transgenic plants with longer roots, stems and siliques, larger rosette leaves, and bigger seeds compared with those of the wild type, indicating the overall promotion effects of *JcFATA* and *JcFATB* on plant growth and development while *JcFATB* had a larger impact. Compositional analysis of seed oil revealed that all fatty acids except 22:0 were significantly increased in the mature seeds of *JcFATA*-transgenic *Arabidopsis* lines, especially unsaturated fatty acids, such as the predominant fatty acids of seed oil, 18:1, 18:2, and 18:3. In the mature seeds of the *JcFATB*-transgenic *Arabidopsis* lines, most fatty acids were increased compared with those in wild type too, especially saturated fatty acids, such as 16:0, 18:0, 20:0, and 22:0. Our results demonstrated the promotion effect of *JcFATA* and *JcFATB* on plant growth and development, and their possible utilization to modify the seed oil composition and content in higher plants.

## 1. Introduction

Plant seed oils are one of the most important food sources for human beings, and are important raw materials for the production of cosmetics, soaps, paints, pharmaceuticals, emulsifiers and lubricants, etc. [1,2]. The rapid development of the world economy has led to an increasing demand for energy and the depletion of petroleum resources. Biodiesel is a kind of processed fuel that is a diesel equivalent from biological sources, such as plant seed oils, and can be used in diesel engine vehicles [3]. The main component of biodiesel is fatty acid methyl esters, which have similar properties to petroleum, and is considered as the main target of new energy alternatives. Although there are still many problems in the process from raw material production to practical application, the feasibility of using vegetable oils to produce bioenergy has been drawing increasing attention [3].

*Jatropha curcas* L. is a perennial woody plant of the Euphorbiaceae family, mainly distributed in sub-tropical or tropical areas [4]. *Jatropha* is widely used in soil reclamation and improvement, and can also be used as the raw materials for industrial production and pharmaceutical compounds, and in many other fields [5,6]. This plant has attracted worldwide attention because of the high oil content in its seeds (up to 50%), which has been considered as a promising biodiesel plant because its seed oil is suitable for biodiesel production [7,8,9,10,11]. At present, there are some problems in the planting and production of *J. curcas*, such as low seed yield and poor oil composition ratio, so it is necessary to genetically improve *J*. *curcas* to develop new varieties with high seed yield and good quality seed oil for biofuel production. Limited by the unclear genetic origin and low genetic diversity of *Jatropha* plants, traditional breeding methods are time-consuming and inefficient, making it difficult to make breakthrough progress [12]. Using genetic engineering technology to improve *Jatropha* can make up for the time-consuming and laborious disadvantages of traditional breeding [13,14]. In order to genetically improve *Jatropha* and obtain superior varieties, more genes that can be used for *Jatropha* improvement need to be identified functionally and their effects on plant growth and development also need to be explored.

Fatty acid synthesis is a very important metabolic process, which plays an important role in plant growth and development. Plant fatty acid synthesis starts in plastid by fatty acid synthase complex (FAS), which usually generates palmitoyl-ACP (16:0-ACP) and stearoyl-ACP (18:0-ACP) through continuous elongation of fatty acid chains in a 2-carbon increase for each cycle [15]. In plants, plastidial acyltransferases can terminate de novo fatty acid synthesis, and the acyl group of acyl-acyl carrier protein (acyl-ACP) can be used to produce glycerolipids in plastids (prokaryotic pathway) or, alternatively, acyl-ACP thioesterases (FATs) can release free fatty acids and ACP by hydrolyzing acyl-ACP [16]. Moreover, the released free fatty acids can be exported to the cytosol and re-esterified to CoA for glycerolipid biosynthesis in the endoplasmic reticulum (eukaryotic pathway) [17]. Therefore, FATs are the key enzymes in de novo synthesis of free fatty acids in higher plant plastids and play an important role in the distribution of de novo-synthesized free fatty acids between the prokaryotic and eukaryotic pathways [18,19,20,21]. Due to the distinct substrate specificity of different FATs, FAT can influence the chain length and saturation degree of fatty acids, and the composition of fatty acids in various organs of higher plants [21,22,23,24]. FAT can be divided into two families, i.e., FATA and FATB, according to their preference for substrates and the differences in their amino acid sequences [25]. In general, FATA can promote the synthesis of unsaturated fatty acids and increase the content of unsaturated fatty acids (mainly oleic acid 18:1), and FATB can promote the synthesis of saturated fatty acids and increase the content of saturated fatty acids (mainly palmitic acid 16:0 and stearic acid 18:0) [26,27].

To date, *FAT* genes in *Arabidopsis thaliana* L. have been studied extensively. The *Arabidopsis* genome contains two *FATA* genes, i.e., *AtFATA1* and *AtFATA2*, and one *FATB* gene *AtFATB* [25]. Overexpression of *AtFATB1* driven by a seed-specific *rapeseed* napin promoter led to greatly increased contents of 16:0, 18:0, and 14:0 fatty acids, whereas unsaturated fatty acids, including 18:1, 18:2, 18:3, 20:1, and 22:1, showed reduced contents compared to the wild type [28]. Downregulation of *AtFATB1* through anti-sense RNA resulted in a reduced content of 16:0 in flowers and mature seeds, but no visible phenotypic changes were observed for the transgenic *AtFATB1*-antisense lines [28]. A knockout T-DNA insertion mutant of *AtFATB* named *fatb-ko* showed severe growth inhibition, abnormal seed morphology, reduced seed viability, and significantly decreased fatty acid contents, especially 16:0 and 18:0, in various organs, indicating *AtFATB* plays an essential role in plant growth and seed development as a main determinant in the synthesis of saturated fatty acids and their derivatives in *Arabidopsis* [29]. They also found that endogenous FATA activity was not increased in the mutant, suggesting no compensation adjustment between FATA and FATB, although previous studies found that the substrate specificities of AtFATA and AtFATB were 18:1 > 18:0 > 16:0 and 16:0 > 18:1 > 18:0, respectively [26,30]. Further study showed that the main metabolic response of the *Arabidopsis* plants to the disruption of *AtFATB* was an increased turnover of fatty acids, including synthesis and degradation, in *fatb-ko* plants [31].

A *fata1 fata2* double mutant, generated by crossing two single mutants with a T-DNA insertion in the promoter region of *AtFATA1* and *AtFATA2*, showed a ca. 60% and 50% decrease of the *AtFATA1* and *AtFATA2* expression levels, respectively, compared with the wild-type *Arabidopsis*. The *fata1 fata2* plants did not show obvious morphological changes, but the seed oil content and fatty acid composition in the dry seeds were affected, including decreased contents of 18:0, 18:1, and 18:2 fatty acids [32]. A T-DNA insertion mutant in the promoter region of *AtFATA2*, with a significantly decreased expression of *AtFATA2*, showed longer siliques, more seeds per silique, slightly small seeds, and increased contents of most types of fatty acids except for 24:0 in dry seeds compared with the wild type, indicating that *AtFATA2* plays an important role in seed lipid metabolism and silique development [33].

In a previous study, *JcFATB1* was isolated from immature seeds of *J*. *curcas* and was strongly expressed in immature seeds detected by semi-quantitative RT-PCR. Ectopic overexpression of *JcFATB1* in *Arabidopsis* driven by a seed-specific promoter could significantly increase the contents of saturated fatty acids, and decrease the contents of unsaturated fatty acids [34]. Dani et al. compared the protein sequences of JcFATA and JcFATB with AtFATA and AtFATB using bioinformatic methods and found three potential conserved catalytic active sites, i.e., the catalytic triad of N, H, and C, in JcFATA and JcFATB protein sequences, but functional verification of these potential active sites has not been reported [35]. The expression profile analysis of some key fatty acid enzyme genes showed that the expression levels of *JcFATA* were increased while the expression of *JcFATB* was decreased with seed development [36]. In this study, the expression patterns of *JcFATA* and *JcFATB* and subcellular localization of JcFATA and JcFATB were firstly analyzed, and their effects on plant growth and development and seed oil contents were further studied by heterologous expression in *A. thaliana*. Their possible roles in seed oil improvement were also discussed.

## 2. Results

### 2.1. JcFATA and JcFATB Encode Typical FAT Proteins Localized in Chloroplasts

*Arabidopsis FATA* and *FATB* genes have been functionally investigated in previous studies [28,29,32,33]. In *Arabidopsis*, there are two *FATA* genes, i.e., *AtFATA1* (At3g25110/BT024746.1) and *AtFATA2* (At4g13050/NM_113415.4), and one *FATB* gene *AtFATB* (At1g08510/BT008505.1) [25,37]. We then used *Arabidopsis FATA* and *FATB* sequences to search the homologous genes of *J. curcas* in the NCBI GenBank database and the published *Jatropha* genome database [38,39,40]. We found that there were 5 *FAT* family genes in the *Jatropha* genome, i.e., *JcFATA*, *JcFATB*, *JcFAT-L1*, *JcFAT-L2*, and *JcFAT-L3* with the NCBI GenBank accession numbers EU267122.2, EU106891.1, JX966081.1, JX966082.1, and JX966083.1, respectively (Appendix A).

The CDS sequences of *JcFATA* and *JcFATB* are 1110 and 1257 bp in length, encoding proteins containing 369 and 418 amino acids, respectively. We searched the *Jatropha* Genome Database [41] using the *JcFATA* and *JcFATB* mRNA sequences to obtain their corresponding genome sequences, and the results showed that the genomic sequences of *JcFATA* and *JcFATB* are 3680 and 3672 bp in length from the start codon to the stop codon, and *JcFATA* contains 6 introns and 7 exons, and *JcFATB* includes 5 introns and 6 exons (Figure 1A). Phylogenetic analysis of 25 FAT family proteins belonging to 10 families of dicotyledonous flowering plants, including *A. thaliana* and *J. curcas*, showed that JcFATA was grouped in the same evolutionary clade as AtFATA1 and AtFATA2 while the remaining four FAT proteins of *J. curcas* were in the same clade as AtFATB. JcFATA showed the highest homology with HbFATA, and JcFATB showed the highest homology with VfFATB, which were all from Euphorbiaceae plants (Figure 1B). These results indicated that *JcFATA* and *JcFATB* might be functional and important *FAT* genes in *J. curcas*, so we conducted further studies on these two genes.

Plant fatty acyl-ACP thioesterase (FAT) proteins are encoded by nuclear genes and targeted to plastids by a transit peptide at the N-terminus. Previous studies have predicted the transit peptide sequences in the N-terminals of JcFATA and JcFATB as shown in Appendix A according to the previous studies [34,35,42,43]. To determine the subcellular localization of JcFATA and JcFATB, the intact ORF without the stop codon of *JcFATA* and *JcFATB* were fused to the N-terminus of GFP (green fluorescent protein), respectively, driven by 35S promoter, and then used to transform *Arabidopsis* green leaf protoplasts for transient expression. In transformed cells, the green GFP signals overlapped with red chlorophyll autofluorescence, which indicated that JcFATA and JcFATB both localized in chloroplasts (Figure 1C).

### 2.2. JcFATA and JcFATB Are Constitutively Expressed Genes with Similar Expression Profiles

A previous study showed that *JcFATA* is a constitutively expressed gene as examined by the RT-PCR and GUS reporter system [44]. In this study, we detected the expression profiles of *JcFATB* using the same materials and method at the same time as the previous study by Liu et al. [44], and we also checked the expression patterns of *JcFATA* and *JcFATB* in developing seeds at different stages. As shown in Figure 2A, *JcFATB* was expressed at relatively low levels in stems, and showed relatively high expression levels in flowers, roots, and leaves, which is very similar to *JcFATA* [44]. During seed development, the expression of *JcFATA* and *JcFATB* increased during early developmental stages (20 to 40 DAF, days after flowering) but reduced during the late stage of seed development (60 DAF), and the highest expression was found in immature seeds at 40 DAF for both genes (Figure 2B).

In order to study the detailed spatiotemporal expression patterns of *JcFATB*, the *GUS* expression vector driven by the *JcFATB* cognate promoter was transformed into wild-type *A. thaliana*, and the screened homozygous transgenic lines were used for GUS histochemical staining. We planted the *JcFATA-GUS* plants used in the previous study [44] and the *JcFATB-GUS* plants generated in this study under the same condition and then performed GUS staining at the same time for comparison. GUS activity was detected in the radicles and cotyledon leaves of 5-day-old seedlings for both genes. The GUS signals were almost undetectable in the whole hypocotyl of the *JcFATA-GUS* plant but only undetectable in the middle part of the hypocotyl of the *JcFATB-GUS* plant (Figure 2(C1) and Appendix A). For 15-day-old *JcFATB-GUS* seedlings, high levels of GUS expression were detected in the whole plant except for hypocotyl (Figure 2(C2)), but GUS activity was almost undetectable in the petioles and hypocotyl of 15-day-old *JcFATA-GUS* seedlings (Appendix A). High GUS staining was detected in the full expanded leaf blades, especially in the leaf veins of 30-day-old plants for both genes (Figure 2(C3) and Appendix A). GUS staining of the inflorescences during the flowering stage showed that almost no GUS activity was detected in the inflorescence rachis while relatively weak and strong GUS signals were detected in flowers during the early developmental stages and late developmental stages, respectively, for both genes (Figure 2(C4) and Appendix A). For *JcFATB-GUS* plants, GUS activity was relatively low in fruit pots during different development stages, and blue signals were mainly found in the tips of the siliques and the junction of the silique base and the peduncle (Figure 2(C5)), which is very similar to a previous study [44]. The results showed that *JcFATB-GUS* plants showed a very similar GUS staining pattern to *JcFATA-GUS* plants, but *JcFATB-GUS* showed stronger GUS staining signals in all the tissues detected (Figure 2C and Appendix A).

### 2.3. Ectopic Expression of JcFATA and JcFATB and Their Mutant Versions Affected the Fatty Acid Accumulation in E. coli

Previous study suggested that there may be 3 possible conserved catalytic active sites, i.e., 265N (asparagine), 267H (histidine), and 302C (cysteine) for JcFATA, and 315N, 317H, and 352C for JcFATB, similar to AtFATA and AtFATB (Appendix A) [35]. In order to verify the catalytic activity of the active sites 302C and 352C in JcFATA and JcFATB, respectively, we constructed the prokaryotic expression vectors for the wild-type *JcFATA* and *JcFATB*, and their mutation versions with a mutation of 302C (TGC, cysteine) to 302F (TTC, phenylalanine) in JcFATA and a mutation of 352C (TGT) to 352F (TTT) in JcFATB, named the JcFATA-Mutation and JcFATB-Mutation, respectively (Appendix A, and Figure 3A,B). Then, each construct and the empty vector were transformed into *E. coli* strain Rosetta to induce exogenous gene expression. The fatty acid composition was then assayed to test the effects of JcFATA, JcFATA-Mutation, JcFATB, and JcFATB-Mutation on the fatty acid accumulation in *E. coli*. Compared with the *E. coli* transformed with pCold I empty vector, the contents of 16:1 and 18:1 unsaturated fatty acids were significantly increased in *E. coli* transformed with pColdI-JcFATA, and the contents of unsaturated fatty acids were significantly reduced in the *E. coli* transformed with pColdI-JcFATA-Mutation compared with pCold I and pColdI-JcFATA (Figure 3C). These results indicated that JcFATA can promote the accumulation of unsaturated fatty acids in *E. coli*, and 302C may be a key amino acid residue of the enzyme activity in regulating the accumulation of unsaturated fatty acids. Compared with *E. coli* transformed with pCold I empty vector, the saturated fatty acid contents of 16:0 and 18:0 were significantly increased in *E. coli* transformed with pColdI-JcFATB. Moreover, the saturated fatty acid contents of 16:0 and 18:0 were significantly reduced in the *E. coli* transformed with pColdI-JcFATB-Mutation compared with pColdI-JcFATB, and only 18:0 was significantly reduced compared with pCold I (Figure 3D). These results indicated that JcFATB can promote the production of bacterial saturated fatty acids, and 352C may be a key amino acid residue of JcFATB for the regulation of saturated fatty acid production.

### 2.4. Ectopic Expression of JcFATA and JcFATB Promotes the Growth and Development of A. thaliana

In order to investigate the effect of *JcFATA* and *JcFATB* on plant growth and fatty acid accumulation, we constructed the overexpression constructs of *JcFATA* and *JcFATB* driven by CaMV 35S promoter, and then the resultant constructs were transformed into *Arabidopsis* individually. Three independent homozygous lines, i.e., JcFATA-1, JcFATA-2, JcFATA-3 for *JcFATA*, and JcFATB-1, JcFATB-2, and JcFATB-3 for *JcFATB*, were obtained by planting and screening with hygromycin from the T3 generation. PCR and semi-quantitative RT-PCR showed the existence of *JcFATA* and *JcFATB* transgene and their successful expression in *Arabidopsis* while the transgene and mRNA of *JcFATA* and *JcFATB* were not detected in wild-type Col-0 (Figure 4A). These homozygous lines were used for further observation and analysis.

We then observed and analyzed some growth parameters during the whole life cycle of wild-type Col-0 plants and the above homozygous *JcFATA* and *JcFATB* ectopic expression lines. At 8 DAT (days after transplanting), both *JcFATA* and *JcFATB* lines showed a significantly increased root length compared with that of the wild type. Specifically, the *JcFATB* lines showed a ca. 30–40% increase. Furthermore, the root lengths of *JcFATB* lines were also significantly longer than the *JcFATA* lines (Figure 4B and Appendix A). Very similar results were also observed for the rosette diameters and the numbers of rosette leaves per plants at 20 DAT, i.e., the *JcFATA* and *JcFATB* lines also showed significantly larger and more rosette leaves compared to the wild-type plants (Figure 4C,D and Appendix A). These results indicate that heterologous expression of *JcFATA* or *JcFATB* can promote plant growth during the vegetative stages of *Arabidopsis*, and *JcFATB* showed a greater growth promotion effect than *JcFATA*.

To identify whether *JcFATA* and *JcFATB* can also improve plant growth and development during the reproductive stages of *Arabidopsis*, the plant height and number of bloomed flowers for the wild-type and transgenic lines at 28 DAT were recorded and analyzed. The average plant heights of *JcFATA* and *JcFATB* lines were significantly increased compared with that in the wild type, and the *JcFATB* lines showed a significant increase compared to the *JcFATA* lines, which is very similar to the results obtained during the vegetative stages (Figure 4D and Appendix A). At 28 DAT, some of the flower buds flowered, and the numbers of bloomed flowers at this time were slightly increased in 1 and 2 lines of the *JcFATA* and *JcFATB* plants, respectively, compared with that in the wild type (Figure 4F and Appendix A). At 50 DAT, we observed that the *JcFATA* and the *JcFATB* lines produced subtly longer siliques (Appendix A), and further measurements showed that the average silique lengths of the *JcFATA* lines were slightly increased compared to the wild type, but the *JcFATB* lines showed significantly longer siliques compared with the wild type (Figure 4G). We then analyzed the phenotypes of the mature seeds of the wild-type, *JcFATA*, and *JcFATB* lines. As shown in Table 1 and Appendix A, both *JcFATA* and *JcFATB* lines produced significantly bigger and heavier seeds, and the lengths, widths, and grain weights of 500 dry seeds were increased significantly in the *JcFATA* and *JcFATB* lines compared with those in the wild type. The grain weight of 500 dry seeds was increased by 14–21% for the *JcFATA* lines and 38–45% for the *JcFATB* lines. Our results indicate that the ectopic expression of *JcFATA* and *JcFATB* can significantly increase the seed weight of *Arabidopsis*, and *JcFATB* showed a greater effect.

### 2.5. Ectopic Expression of JcFATA and JcFATB Affects Seed Fatty Acid Accumulation in Arabidopsis

Due to the high expression of *JcFATA* and *JcFATB* in *J. curcus* seeds at 40 DAF and the considerable phenotypical changes in the seeds of the *JcFATA* and *JcFATB* ectopic expression lines, we further examined the fatty acid composition of seed storage lipids in wild-type and ectopic expression lines of *Arabidopsis*. Compositional analyses of seed oil revealed that, except 20:1, the contents of other unsaturated fatty acids, including 16:1, 18:1, 18:2, 18:3, and 22:1, were significantly increased in the dry seeds of *JcFATA* lines compared with those in the wild type (Figure 5A). Moreover, the contents of 18:1, 18:2, and 18:3 were increased by 69–95%, 58–68%, and 52–70%, respectively, in the *JcFATA* lines compared with the wild type (Appendix A). Notably, the contents of the 18:0 and 20:0 saturated fatty acids were also significantly increased in the *JcFATA* lines. These results suggested that *JcFATA* can significantly promote the accumulation of fatty acids, especially the accumulation of unsaturated fatty acids. Similarly, most of the fatty acids were increased in the dry seeds of the *JcFATB* lines compared with those in the wild type except that 22:1 was decreased significantly (Figure 5B). The saturated fatty acids 16:0, 18:0, 20:0, and 22:0 were significantly increased by 84–108%, 124–145%, 65–92%, and 96–170%, respectively, in the *JcFATB* lines compared with the wild type (Appendix A). Regarding the unsaturated fatty acids, 18:1, 18:2, and 18:3 were also significantly increased at relatively lower levels than the saturated fatty acids. The significantly increased contents of fatty acids, especially the saturated fatty acids, in the mature seeds of *A. thaliana* indicate that *JcFATB* has a significant stimulative effect on the accumulation of fatty acids, especially on the accumulation of saturated fatty acids. As seen from the data shown in Appendix A, *JcFATA* showed a stronger promoting effect on the accumulation of unsaturated fatty acids than *JcFATB* while *JcFATB* showed a stronger promoting effect on the accumulation of saturated fatty acids than *JcFATA*.

To deduce whether the seed oil contents were altered in the transgenic *Arabidopsis* lines expressing *JcFATA* and *JcFATB* compared with the wild-type Col-0, we calculated the percentages of the fatty acids presented in Figure 5 in the total seed oils by analyzing the fatty acid GC-MS (gas chromatograph-mass spectrometer) data. Surprisingly, the percentages of almost all the saturated fatty acids and unsaturated fatty acids did not change significantly in the *JcFATA* lines compared with those in the wild type, and the *JcFAT**B* lines only showed significantly increased percentages for 18:0 and a significant decrease for 22:1 (Appendix A). Moreover, the total percentages were evidently increased and decreased for the saturated fatty acids and the unsaturated fatty acids, respectively, in the *JcFATB* lines. However, the total percentages for the saturated and unsaturated fatty acids did not change much in both the *JcFATA* lines and *JcFATB* lines (Appendix A). We also calculated the GC-MS peak ratios of the total saturated and unsaturated fatty acids with the standard substance (ethyl decanoate), and we found that the ratios were increased significantly for both the *JcFATA* lines and *JcFATB* lines compared with that in the wild type (Appendix A). These data suggest that the seed oil content might be increased in the *Arabidopsis* line with ectopic expression of *JcFATA* and *JcFATB*. To prove this, further experiments are required.

## 3. Discussion

Acyl-ACP thioesterases (FATs) catalyze the termination of the FAS cycle, and the genes encoding FATs have been cloned from many plants, such as safflower [42], oil seed rape [45], sunflower [42,46,47], oil palm [48], and *A. thaliana* [25]. In vitro enzymatic activity experiments showed that FATA had the highest catalytic activity on unsaturated fatty acid 18:1-ACP followed by 18:0-ACP, 16:1-ACP, and 16:0-ACP. FATB had the highest catalytic activity on saturated fatty acid 16:0-ACP followed by 18:1-ACP, 18:0-ACP, and 16:1-ACP [26,27].

A previous study showed that the contents of the total seed oil and the four major fatty acids, i.e., palmitic acid (16:0), stearic acid (18:0), oleic acid (18:1), and linoleic acid (18:2), varied significantly in 19 different accessions of *J. curcas* [49], which indicated that the seed oil content and composition could be improved through conventional breeding and biotechnology. Furthermore, *Jatropha* seed oil contains a high content of polyunsaturated fatty acids, which may reduce the oxidation stability of the oil, and can also lead to an increase in nitrogen oxide emissions after burning [50,51]. The ideal biodiesel should contain more monounsaturated fatty acids, such as oleic acid (18:1), rather than polyunsaturated fatty acids, such as linoleic acid (18:2) and linolenic acid (18:3), so the composition of fatty acids in seed oil has yet to be improved [52]. FATs are specific for substrate selection and have a certain determining effect on the types of fatty acids produced in seeds and can be used as potential target genes for the improvement of seed oil in plants. Several studies have showed that FATA can increase the accumulation of oleic acid (18:1) and linoleic acid (18:2) in several plant species [26,53,54,55].

FATs are expected to localize in plastids or chloroplasts, and the previous study showed that AtFATA2 localized in chloroplasts [33]. In this study, we found that JcFATA and JcFATB both localized in chloroplasts as detected by transient expression in *Arabidopsis* leaf protoplasts, indicating they may have similar functions to their *Arabidopsis* homologs. However, we also found that the fluorescence signal patterns of JcFATA and JcFATB were not identical, and the fluorescence signals for JcFATA-GFP showed an uneven pattern, which is very similar to AtFATA2, while the signals of JcFATB-GFP were more contiguous in a single chloroplast. Previous studies indicated that the proteins localized in the outer or inner membrane of chloroplasts usually show uneven fluorescence signals [33,56,57,58]. However, the GFP fluorescence signals of thylakoid proteins are usually uniformly distributed throughout the chloroplast [58]. The difference in their localization in chloroplasts suggests that FATA and FATB may reside in different positions to exert their hydrolysis function [33,57,58,59,60].

*FATB* in *Cuphea hyssopifolia* is widely expressed in the developing embryos [61]. *AtFATB* is widely expressed in all organs but has the highest expression level in flower organs [28]. *JcFATB* transcripts were detected in all the organs examined, with the highest expression in the developing seeds at 32 DAF [34]. The expression of *JcFATA* was increased with the development of the endosperm and then decreased during seed maturation [62]. In this study, we found that the spatiotemporal expression patterns of *JcFATA* and *JcFATB* were similar to those of *FATA* and *FATB* in *Arabidopsis* and other plants, suggesting that JcFATA and JcFATB may act as main acyl-ACP thioesterases and share similar functions in the various tissues and developmental stages of *Jatropha*, especially in fatty acid synthesis. This may, in turn, make FATs essential for plant viability by affecting fatty acid metabolism as shown by previous studies [32,37]. In the present study, longer and bigger seeds were produced by *Arabidopsis* plants with ectopically expressed *JcFATA* or *JcFATB* driven by CaMV 35S promoter, and the increased fatty acid contents and changed seed oil compositions in mature seeds may in turn affect seed development. Previous research and the present study suggest that *JcFATA* and *JcFATB* may play an important role in seed development by participating in fatty acid synthesis [34].

Prokaryotic expression studies showed that plant FATs could influence the fatty acid accumulation of *E. coli*. Ectopic expression of a plant *FAT* gene encoding a medium-chain-specific FAT, named BTE from *Umbellularia californica*, in both a normal and a fatty acid synthesis-deficient mutant of *E. coli* had a limited impact on the normal strain and a huge impact on the mutant strain on the fatty acid composition [63]. *E. coli* expressing *CsFATA* of *Coriandrum sativum* showed an increased unsaturated fatty acid content, indicating that *CsFATA* plays a certain role in promoting the formation of unsaturated fatty acids [30]. *E. coli* can also be used for the production of free fatty acids by blocking the fatty acid elongation process caused by ectopic expression of *FAT* genes from other species [64]. A previous study found that ectopic expression of seven exogenous *FAT* genes, including *AtFATA*, in *E. coli* led to significantly increased production of free fatty acids, and the fatty acid compositions of the strains with different *FATs* showed substrate specificity, suggesting that *FATs* can be engineered and introduced into *E. coli* to produce free fatty acids [65]. Ectopic expression of a *FatB* gene cloned from *Diploknema butyracea* (*Madhuca butyracea*) in *E. coli* caused a significant increase in the content of 16:0 saturated fatty acids in the supernatant [66]. In this study, we found that ectopic expression of *JcFATA* and *JcFATB* in *E. coli* led to significantly increased accumulation of the unsaturated fatty acids 16:1 and 18:1, and saturated fatty acids 16:0 and 18:0, and the mutated version of *JcFATA* and *JcFATB* in 1 of the 3 possible conserved catalytic active sites caused decreased accumulation of the unsaturated fatty acids 16:1 and 18:1, and decreased accumulation of the saturated fatty acids 16:0 and 18:0 (Figure 3). These results suggest the possible role of 302C (cysteine) in JcFATA, and 352C in JcFATB for their enzymatic activity. In bacteria, the synthesis of cyclopropane fatty acid (CFA) on the cell membrane plays an important role in its adaptation to the changing environment and response to various environmental stresses. Ectopic expression of peanut (*Arachis hypogaea*) *AhFat**A* in *E. coli* affected the fatty acid compositions of the membrane lipid, leading to significantly increased accumulation of 16:1 and 18:1, and affecting bacteria growth [66]. Ectopic expression of *JcFATA* or *JcFATB* might have the same effect on the CFA synthesis of the cell membrane in *E. coli*, such as for *AhFatA* [67].

As the main FATs, FATA and FATB are required for the final step of de novo fatty acid biosynthesis, which determines the metabolic flux of fatty acid metabolism in plants, and are therefore essential for plant survival [67]. Our results showed that the ectopic expression of *JcFATA* and *JcFATB* produced longer roots and siliques, larger and more rosette leaves, more flowering buds, and greater plant height, and the *JcFATB* lines showed greater increases in these phenotypes compared with the *JcFATA* lines (Figure 4). Corresponding to the stimulative effects on plant growth, we found that most of the fatty acids of the mature seeds were increased due to the ectopic expression of *JcFATA* and *JcFATB*, and the significant increase in some fatty acids, such as 18:0, 18:1, 18:2, and 18:3 for the *JcFATA* lines and 16:0, 18:0, and 20:0 for the *JcFATB* lines, might be responsible for the plant growth promotion (Figure 5). The biosynthesis and supply of saturated fatty acids and unsaturated fatty acids are essential in plant growth and development [29]. Although we did not measure the fatty acid contents in other organs, we expect their fatty acid contents would be increased similar to that observed in mature seeds considering their phenotypic changes. The phenotypic changes in the transgenic *Arabidopsis* plants in this study may be due to metabolic responses to the changed fatty acid contents and compositions. Protein S-acylation, especially palmitoylation, is an important post-translational modification, which is essential for the regulation of activity and localization of membrane-related signaling proteins [68]. As a reversible modification of the cysteine residues of target proteins, protein S-acylation plays important roles in multiple aspects of protein function, such as localization, stability, trafficking, and conformation. The increased contents of 16:0 and other fatty acids may affect signal transduction in multiple growth and development processes by promoting protein S-acylation [69].

One interesting work showed that overexpression of *Arabidopsis* ceramide synthase genes *LOH1*, *LOH2*, and *LOH3* under the control of the CaMV 35S promoter in Col-0 led to differentially altered growth and extensive changes in sphingolipid metabolism, in which *LOH1* and *LOH3* overexpression lines showed a significant increase in plant size and biomass with little alteration in the sphingolipid composition or content on a tissue mass basis. However, *LOH2* overexpression led to severely dwarfed plants with differentially altered sphingolipid profiles in their rosettes compared with the wild type [70]. Moreover, an interesting study in cotton and *Arabidopsis* found that the very-long-chain fatty acids (VLCFAs) with saturated fatty acids, especially C24:0, play an important role in cell elongation and expansion by activating ethylene biosynthesis and signaling. Comprehensive lipid analysis indicated that linolenic (18:3) and palmitic (16:0) acid are the most abundant fatty acids in the development of cotton fibers [71]. The significant decrease in 22:1 in *JcFATB* ectopic *Arabidopsis* lines could be caused by the increase in the accumulation of C16 and C18 fatty acids caused by *JcFATA* overexpression, and determination of the VLCFAs content might explain this question in our further study. Overexpression of *AtFAAH*, which encodes the fatty acid amide hydrolase that is responsible for hydrolyzing *N*-acylethanolamines, a group of fatty acid derivatives of ethanolamine, into ethanolamine and their corresponding free fatty acids, under the control of CaMV 35S promoter, showed a significant promotion effect on plant growth [72]. From these studies, we could speculate that the overexpression of some genes related to fatty acid metabolism or related pathways may promote plant growth and development.

Vegetable oils and their derivatives have important economic value and have been used as important industrial raw materials and nutritional sources. Thus, significant attention has been given to the yield and quality of seed oils. In the effort to improve seed oil, several genes have been used in previous studies; however, with an improved seed oil content, some genes showed inhibition effects on plant growth and development [73,74]. Ectopic expression of a fatty acid dehydrogenase gene increased the content of α-linolenic acid (18:3) in soybean seeds, but this was accompanied by a serious decrease in the total oil content [75,76]. Several studies showed that FATs can impact on the production of glycerolipds by affecting the export of acyl moieties to ER in *Arabidopsis* [28,29,31,32,33]. Although *JcFATA* and *JcFATB* had different effects on the fatty acid composition of seed oils when ectopically expressed in *A. thaliana*, these two genes can promote the growth and development of plants, with almost the same effects on various organs. Our results suggested that *FAT* genes are potential genes that can be used for seed oil improvement, and the *FAT* genes from other plant species could also be used. In this study, we used the constitutive CaMV 35S promoter to drive the expression of *JcFATA* and *JcFATB* instead of the seed-specific promoter used in other studies, and we also obtained very promising results. Increased unsaturated fatty acids may promote the chilling tolerance of plants as described previously, which suggests that *FATA* might be used to improve cold tolerance and modify the seed oil composition, similar to other genes involved in fatty acid synthesis [75,76,77,78,79,80,81,82]. Overexpression of *JcFATA* may provide a possible potential to achieve a combined increase in unsaturated fatty acids, seed yield, and chilling tolerance in *J. curcas*. Simultaneous overexpression of *FATA* and *FATB* with different expression levels might also be a strategy for modifying the seed oil content in different plants.

## 4. Materials and Methods

### 4.1. Plant Materials and Growth Conditions

For *J. curcas*, a genotype named M-19 collected from Yunnan Province of China was used in this study [83,84]. The mature tree plants were grown on the farm of South China Agricultural University in Guangzhou, China. *Arabidopsis thaliana* L. ecotype Columbia (Col-0) was used in this study. The wild-type seeds of Col-0 were sterilized in 2% sodium hypochlorite for 15 min, and then rinsed 3 times with sterile water and inoculated on 1/2 MS medium [85]. Seeds from transgenic plants were planted on the same basic medium containing 50 mg/L hygromycin. After treatment in darkness at 4 °C for 48 h, the seeds were germinated at 22 °C under a 16 h day/8 h night photoperiod with a light intensity of 120 μmol/m^2^ s and 80% relative humidity in an artificially controlled plant growth chamber. The seedlings were transferred to soil 7 days after germination and grown under the same growth conditions. All mediums were supplemented with 100 mg/L myo-inositol (Sigma-Aldrich, St. Louis, MO, USA) and 2.5% sucrose, adjusted to pH 5.8–6.0, and solidified with 0.6% agar prior to autoclaving at 1.4 kg cm^−2^ for 20 min.

### 4.2. Phylogenetic Analysis of FAT Proteins

The protein and mRNA sequences of FATA and FATB from *J. curcas* and *A. thaliana* were downloaded from National Center for Biotechnology Information [86] and the NCBI accession numbers for each sequence used in this study are listed in Appendix A. BLAST analysis was performed to search the homologous proteins in other species using JcFATA and JcFATB as the query, respectively. In total, 12 FATAs and 6 FATBs from other dicotyledonous plant species were chosen and used for phylogenetic analysis, including *Camelina sativa* FATA (AFQ60946.1), *Brassica napus* FATA1 (XP013689229.1), *Citrus sinensis* FATA1 (KAH9728754.1), *Hevea brasiliensis* FATA1 (XP021663762.1), *Morus notabilis* FATA1 (XP010104178.1), *Carya illinoinensis* FATA1 (XP042978187.1), *Glycine max* FATA1 (XP006602508.1), *Vitis vinifera* FATA1 (XP010646566.1), *Prosopis alba* FATA1 (XP028766924.1), *Carica papaya* FATA1 (XP021909479.1), *Cucurbita maxima* FATA1 (XP023004726.1), *Glycine max* FATB (NP001237802.2), *Arachis hypogaea* FATB (ABO38558.1), *Vernicia fordii* FATB (AHI86053.1), *Hevea brasiliensis* FATB (XP021672870.1), Populus tomentosa FATB (ABC47311.1), and *Salix suchowensis* FATB (KAG5225597.1). Multiple alignments of the protein sequences were generated with Clustal X [87]. A phylogenetic tree was built by the neighbor-joining method with MEGA 5.05 using 1000 bootstrap replicates [88].

### 4.3. RNA and cDNA Preparation

Total RNA of the flowers after 5 DAF (days after flowering); roots, leaves, and stems from mature plants; seeds at different developmental stages, i.e., 20, 40, and 60 DAF of *J. curcas*; and 30-day-old rosette leaves of *A. thaliana* was extracted using a Plant RNA Kit (Solar Technologies, Gaithersburg, MD, USA) according to the manufacturer’s protocol. Total RNA (1 μg) free of DNA was used for cDNA synthesis using M-MLV Reverse Transcriptase (Promega, Madison, WI, USA) according to the manufacturer’s protocol.

### 4.4. Semi-Quantitative RT-PCR

The primer pairs specific for *JcFATA* and *JcFATB* were designed using Primer Premier 5.0 software (PREMIER Biosoft, San Francisco, CA, USA), and validated to generate single PCR products with the expected sizes. For analysis of the expression profiles of *JcFATA* and *JcFATB* in *J. curcas* by semi-quantitative RT-PCR, 1 μL of a 1:1 diluted RT reaction product was used as template in a 20 μL reaction volume with primer pairs JcFATA-RTSF and JcFATA-RTSR to amplify a 376 bp coding sequence of *JcFATA* and JcFATB-RTSF and JcFATB-RTSR to amplify a 324 bp coding fragment of *JcFATB*. The house-keeping gene *JcActin7* (XM_012232498.2) was used as the reference gene. For amplification, the primers JcACT7-SF and JcACT7-SR were used, which can amplify a 393 bp PCR fragment. For analysis of the expression of *JcFATA* and *JcFATB* in *Arabidopsis* by semi-quantitative RT-PCR, the same primer pairs and RT-PCR reaction system were used, except that the constitutive expression gene *AtActin2* (NM_112764.4) was used as the internal control, and the primers AtACT2-F and AtACT2-R were used for amplification, which amplified a 191 bp PCR fragment. All the primers used in this study are listed in Appendix A.

### 4.5. Histochemical GUS Assay

A modified pCAMBIA1300 vector with the CaMV 35S promoter, *GUS* gene, and *NOS* terminator in the multiple cloning sites was used for *GUS* fusion construction [89]. A 2234 bp genomic sequence upstream of the ATG start codon of *JcFATB* was amplified from M-19 by PCR using the primer pairs JcFATB-GUS-F and JcFATB-GUS-R. Then, the CaMV 35S promoter of the above pCAMBIA1300-GUS vector was replaced as described for the construction of the *JcFATA-GUS* fusion in which a 2271 bp promoter region of *JcFATA* was used [44]. The resulting construct was transformed into *Agrobacterium tumefaciens* stain EHA105 by the freeze-thaw method [90]. Transformation of *A. thaliana* Col-0 plants was performed using the floral dip method as previously described [91]. The whole seedlings or tissue cuttings of the wild-type and transgenic Col-0 plants at different developmental stages were stained in 2 or 5 mL tubes. GUS staining was performed as described previously [92]. After staining and decoloration, samples were observed and photographed with a stereomicroscope LeicaMZ16 (Leica, Wetzlar, Germany).

### 4.6. Subcellular Localization Analysis

The intact coding region sequences of *JcFATA* and *JcFATB* without a stop codon were amplified by RT-PCR with the primer pairs of JcFATA-PGFP-F and JcFATA-PGFP-R and JcFATB-PGFP-F and JcFATB-PGFP-R, and inserted into the transient expression vector pUC18-35S-eGFP between the CaMV 35S promoter and the *GFP* (*green fluorescent protein*) gene, generating an in-frame fusion for each gene. The leaves of 8-day-old *Arabidopsis* seedlings were cut into 1–2 mm pieces using a fresh sharp blade and used for protoplast preparation. Protoplasts were quantified using a hemocytometer under a microscope, and the fusion constructs for each gene and the control expression vector was introduced into the protoplasts as described [93]. GFP signals were observed under a fluorescence microscope OLYMPUS MF30 (Olympus Corporation, Tokyo, Japna) with the excitation and emission filters Ex480 ± 20/DM505/BA535 ± 25 and Ex535 ± 25/-DM565/BA645 ± 37.5 for GFP and chlorophyll auto-fluorescence, respectively. All fluorescence images obtained were processed with LSM 5 Image Browser (Carl Zeiss AG, Oberkochen, Germany).

### 4.7. Construction of JcFATA and JcFATB Site-Directed Mutagenesis Vectors

The prokaryotic expression vectors for *JcFATB* (pColdI-JcFATB) were constructed first as previously described for *JcFATA* (pColdI-JcFATA), which was also used in this study [94]. The coding region sequence of *JcFATB* without a stop codon was amplified by RT-PCR with the primer pair JcFATB-PC-F and JcFATB-PC-R, and then inserted into the prokaryotic expression vector pCold I (TaKaRa, Dalian, China). Site-directed mutagenesis vectors for *JcFATA* and *JcFATB* were constructed using a QuickChange Lighting Site-Directed Mutagenesis Kit (Stratagene, La Jolla, CA, USA) according to the protocol provided by the manufacturer. Briefly, the recombinant plasmids of pColdI-JcFATA and pColdI-JcFATB were used as the PCR templates, and the mutagenic primers JcFATA-PC-FM and JcFATA-PC-RM for *JcFATA* and JcFATB-PC-FM and JcFATB-PC-RM for *JcFATB* containing the designed mutations were used to amplify and introduce mutations. The resulting vectors were named pColdI-JcFATA-Mutation and pColdI-JcFATB-Mutation.

### 4.8. Analysis of the Fatty Acid Composition of E. coli

To induce *JcFATA* and *JcFATB* expression in *E. coli*, the strain Rosseta (DE3) (TransGen Biotech, Beijing, China) was transformed with pCold I, pColdI-JcFATA, pColdI-JcFATA-Mutation, pColdI-JcFATB, and pColdI-JcFATB-Mutation, respectively. Cultures were grown at 15 °C to OD_600_ of 0.6, and then 0.5 mmol/L of IPTG were added to induce the expression of cloned genes. The cells were then collected by a 10-min centrifugation at 4000 rpm, and then resuspended in 5 mL of ddH_2_O. After a 10-min centrifugation at 4000 rpm, 2 mL of NaOH-methanol solution were added followed by incubation in a 100 °C water bath for 40 min. the solution was then left to cool to room temperature. Two volumes of HCl-methanol solution (80 °C) were added into the tube, incubated in an 80 °C water bath for 40 min with shaking at 80 rpm, and then rapidly cooled to 25 °C in an ice box. Then, 1 mL of n-hexane (Sigma-Aldrich, St. Louis, MO, USA) was added into the cell mixture solution and vortexed for 2 min. After a 5-min centrifugation at 4000 rpm, the upper supernatant was transferred to a new tube. After N_2_ evaporation, FAMEs were dissolved and assayed according to the method used for mature seeds in this study.

### 4.9. Construction of the Overexpression Vectors of JcFATA and JcFATB and Arabidopsis Transformation

The coding sequence (1110 bp) of *JcFATA* and the coding sequence (1257 bp) of *JcFATB* were amplified by RT-PCR with the primer pairs JcFATA-F and JcFATA-R, and JcFATB-F and JcFATB-R, and the PCR products were gel-purified, digested, and inserted downstream of the CaMV 35S promoter cloned into the binary vector pCAMBIA1390. The two recombinant vectors pCAMBIA1390-35S-JcFATA and pCAMBIA1390-35S-JcFATB were transformed into *A. tumefaciens* stain EHA105, and then introduced into *A. thaliana* Col-0 to obtain the transgenic *Arabidopsis* lines ectopically expressing *JcFATA* or *JcFATB* via the floral dip method [91].

### 4.10. Phenotypic Observation and Analysis

For phenotypic observation of the wild-type and transgenic *Arabidopsis* plants, we obtained three independent homozygous lines for *Arabidopsis* plants ectopically overexpressing both *JcFATA* and *JcFATB* by screening the survival rate of the T3 generation in 1/2 MS solid medium containing 50 mg/L hygromycin. After screening, the homozygous lines and the wild-type plants were planted and grown on the same medium without hygromycin under the same conditions. After 7 days, some of the plants were transplanted onto new medium to measure the root length, and all other plants were transplanted into small pots with sterilized nutritional soil and vermiculite at a 1:1 volume ratio. Four plants were planted in each pot. The phenotypes, including the root length, rosette diameter, rosette number, plant height, flowering efficiency, silique length, seed number, seed size, and seed weight, of the transgenic plants and wild-type plants were observed and recorded at 8, 20, 28, and 50 DAT. Regarding the rosette diameter, the largest pair of rosette leaves was used, and the number of rosette leaves was also recorded at 20 DAT. At 28 DAT, the plant height and the flowering efficiency were assayed. The flowering efficiency was designated as the percentage of bloomed flower buds in all the flower buds, i.e., the bloomed ratios. The silique length was measured at 50 DAT, and 10 siliques on the middle part of the main stem for each plant were used. The seed length and width were measured with a micrometer under a microscope, and 50 seeds were measured for each replicate and 3 replicates were used for each line. For each trait, 10 plants were used as 1 replicate and 3 replicates were analyzed in total.

### 4.11. Fatty Acid Analysis of the Mature Seeds of Arabidopsis

Fatty acid methyl esters (FAMEs) were prepared from dry seeds of the wild-type control and the transgenic lines as described previously with minor modification [33]. For each sample, 10 mg of dry seeds were transferred into a 5 mL centrifuge tube, and 1 mL of 5% methanol-sulfuric acid solution and 25 μL of 0.2% dibutyl hydroxytoluene dissolved in methanol were added. Then, the tubes were placed in a 90 °C water bath for 90 min. After incubation, the tubes with samples were placed on ice, and 2 mL of n-hexane and 1.5 mL of 0.9% NaCl were added into each tube. Then, the upper liquid was transferred to a new tube. After N_2_ evaporation, FAMEs were dissolved in 200 μL of n-hexane with 0.216 ng of ethyl decanoate (Sigma-Aldrich, St. Louis, MO, USA) as an internal standard and transferred to a GC vial. The samples were filtered by a 0.22 μm filter (organic phase) and assayed by GC-MS. FAMEs were separated using a 30 m + 0.25 mm DB-23 capillary column with helium as the carrying gas in an Agilent Technologies 7890A Gas Chromatograph, and detected by a flame ionization detector at 230 °C. The program was 170 °C for 1 min followed by an increase of 4 °C/min to 250 °C, which was maintained for a further 3 min, and the column flow was 7.4 mL/min at a split vent ratio of 10:1. FAMEs were identified by comparison with the retention times of reference standards. FAMEs were quantified by comparing the areas of major peaks with those of internal standards.

## Figures and Tables

**Figure 1 ijms-23-04209-f001:**
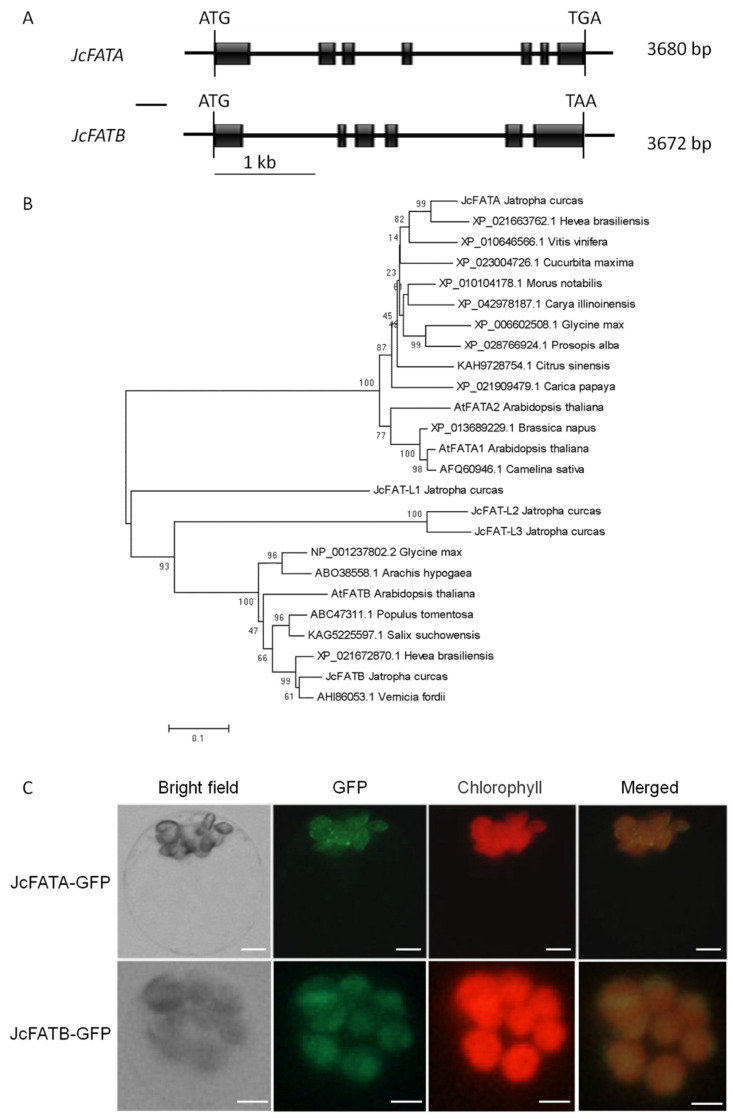
Schemes of the *JcFATA* and *JcFATB* genes and characterization of the JcFATA and JcFATB proteins. (**A**) Genomic organization of the *JcFATA* and *JcFATB* genes. The closed black boxes indicate exons, and connecting lines indicate introns. The ATG start codon and TGA/TAA stop codon are also indicated. (**B**) Phylogenetic tree of JcFATA, JcFATB, and other FAT (fatty acyl-ACP thioesterase) proteins. The coding region sequences were aligned using Clustal W, and the evolutionary relationship was analyzed using the neighbor-joining method. Numbers on branches indicate the percentage of replicate trees in which the associated sequences clustered together in the bootstrap test (1000 replicates). The segment under the phylogenic tree is the evolutionary distance, which was computed using the Poisson correction method. The NCBI accession numbers for the FAT proteins of *A. thaliana* and *J. curcas* are presented in Appendix A. (**C**) Subcellular localization of the JcFATA-GFP and JcFATB-GFP fusion protein in leaf protoplasts isolated from 8-day-old seedlings of wild-type *A. thaliana* Col-0. GFP, green fluorescent protein. Scale bars = 2 μm.

**Figure 2 ijms-23-04209-f002:**
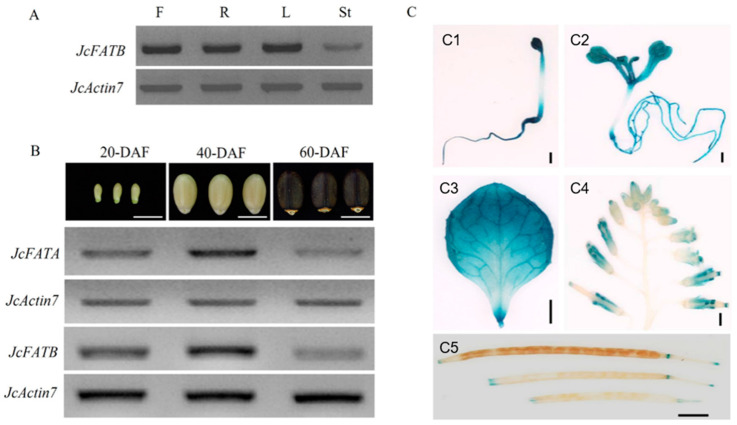
Expression profiles of *JcFATA* and *JcFATB*. (**A**) Expression of *JcFATB* in different tissues analyzed by semi-quantitative RT-PCR in *J. curcas*. F, flowers at 5 DAF (days after flowering); R, L, and St indicates mature roots, leaves, and stems. (**B**) Expression of *JcFATA* and *JcFATB* in developing seeds at 20, 40, and 60 DAF in *J. curcas* as analyzed by semi-quantitative RT-PCR. *Jc**Actin7* was used as an internal control for RT-PCR. Scale bars = 1 cm. (**C**) Expression profiles of *JcFATB* analyzed by GUS staining. (**C1**,**C2**) *JcFATB*:*GUS* seedlings at 5 and 15 DAG (days after germination); (**C3**) A true leaf from a 30-DAG *JcFATB*:*GUS* plant; (**C4**) The top part of inflorescence of the *JcFATB*:GUS plant; (**C5**) Siliques at different development stages of the *JcFATB*:GUS plant. Scale bars = 1 mm.

**Figure 3 ijms-23-04209-f003:**
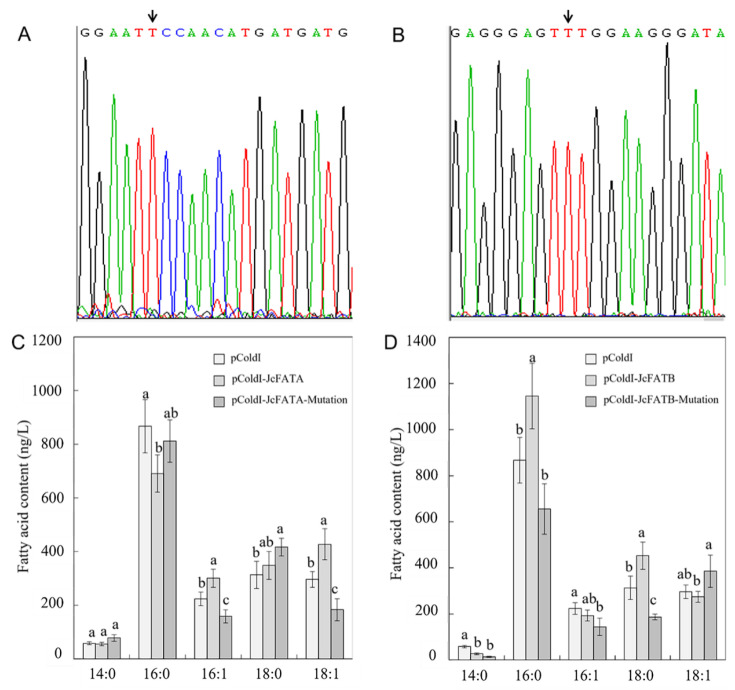
Analysis of the conserved catalytic active sites of JcFATA and JcFATB by prokaryotic expression. (**A**,**B**) Determination of the site-specific mutagenesis of *JcFATA* (**A**) and *JcFATB* (**B**), where the small black arrow indicates the mutated nucleotides and their sequencing peaks. (**C**,**D**) Analysis of the fatty acid contents of *E. coli* clones expressing wild-type *JcFATA* and *JcFATB*, and their mutant versions (JcFATA-Mutation and JcFATB-Mutation), respectively. pCold I was used as the prokaryotic expression vector. Each value in (**C**,**D**) was the mean of three measurements. Error bars = SD (standard deviation). Data within a column followed by different letters (a, b, and c) are significantly different at *p* ≤ 0.05 as determined by Duncan’s multiple range test.

**Figure 4 ijms-23-04209-f004:**
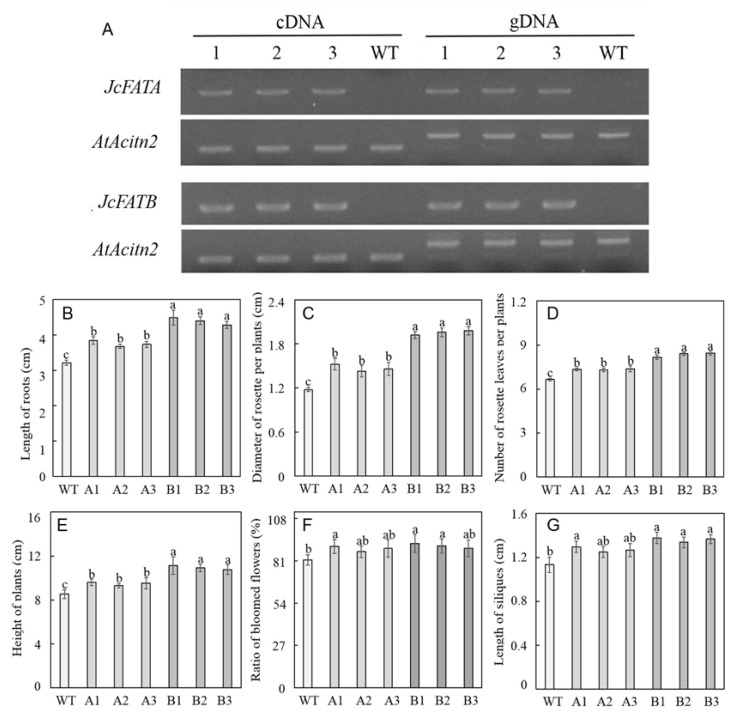
Heterologous expression of *JcFATA* and *JcFATB* promoted plant growth and development of *A. thaliana.* (**A**) Molecular characterization of 3 *JcFATA* (1, 2, 3) and 3 *JcFATB* (1, 2, 3) ectopic expression lines by semi-quantitative RT-PCR and PCR analysis. Wild-type ecotype Columbia (Col-0) was used as the negative control. (**B**) Root length of the 8-DAT (days after transplanting) *A. thaliana* plants expressing *JcFATA* and *JcFATB*. (**C,D**) Diameter and number of rosette leaves of the 20-DAT *A. thaliana* plants expressing *JcFATA* and *JcFATB*. (**E**,**F**) Plant height and bloomed ratio of the 28-DAT transgenic *A. thaliana* plants expressing *JcFATA* and *JcFATB*. (**G**) Silique length of the 50-DAT *A. thaliana* plants expressing *JcFATA* and *JcFATB*. Each value was the mean of three measurements. Error bars = SD (standard deviation). Data within a column followed by different letters (a, b, and c) are significantly different at *p* ≤ 0.05 as determined by Duncan’s multiple range test.

**Figure 5 ijms-23-04209-f005:**
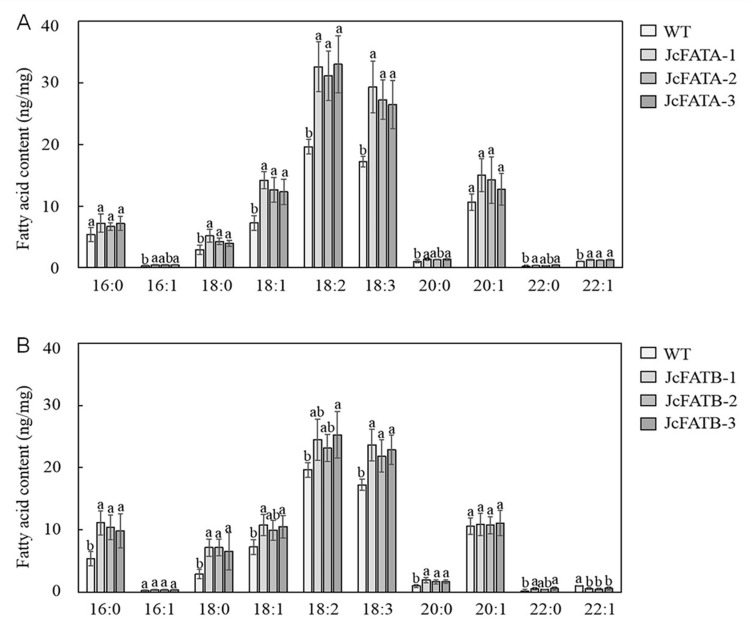
Fatty acid compositions of the seed storage lipids of the *A. thaliana* plants expressing *JcFATA* and *JcFATB*. Fatty acid compositions of the seed storage lipids of the *JcFATA* (**A**) and *JcFATB* (**B**) heterologous expression lines. Each value is the mean of three measurements. Error bars = SD (standard deviation). Data within a column followed by different letters (a, b) are significantly different at *p* ≤ 0.05 as determined by Duncan’s multiple range test.

**Table 1 ijms-23-04209-t001:** Mature seed size and weight of the *JcFATA and JcFATB* ectopic expression lines of *A. thaliana*.

Plant Line	Seed Length (μm)	Seeds Width (μm)	Grain Weight (mg)
WT	475.53 ± 3.01 c	282.47 ± 2.93 c	9.87 ± 0.65 c
JcFATA-1	491.41 ± 7.32 b	298.63 ± 5.26 b	11.53 ± 0.78 b
JcFATA-2	488.18 ± 4.35 b	292.90 ± 1.15 b	11.27 ± 0.55 b
JcFATA-3	494.68 ± 3.83 b	295.96 ± 3.13 b	11.97 ± 0.42 b
JcFATB-1	528.69 ± 7.63 a	310.43 ± 2.55 a	14.23 ± 0.70 a
JcFATB-2	522.88 ± 12.86 a	308.53 ± 3.11 a	13.67 ± 0.51 a
JcFATB-3	527.24 ± 9.78 a	308.16 ± 2.16 a	14.37 ± 0.45 a

Values represent means ± SD (standard deviation), and 50 seeds for grain size and 500 seeds for grain weight were measured for each replicate and 3 replicates were used for each line. Data in the same column followed by different letters (a, b, and c) are significantly different at the *p* ≤ 5% level as determined by Duncan’s multiple range test. For each trait, 10 plants were used as 1 replicate.

## Data Availability

The data presented in this study are available on request from the corresponding author.

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
