# Peer review of "Heterologous Expression of Jatropha curcas Fatty Acyl-ACP Thioesterase A (JcFATA) and B (JcFATB) Affects Fatty Acid Accumulation and Promotes Plant Growth and Development in Arabidopsis"

_ijms, 2022, doi:10.3390/ijms23084209_

Round 1

Reviewer 1 Report

This manuscript deals with acyl-ACP thioesterases from Jatropha curcas, an oilseed of oleochemical and biotechnological interest. Thioesterases or FATs are very important enzymes in lipid synthesis and determine the flow of fatty acids exported from the plastid to the endoplasmic reticulum, where glycerolipids are synthesised.

Part by part, the introduction of the manuscript is correct as it explains well the precedents, objectives and was supported by appropriate references. The methods were correct and well described and the conclusions and discussion were in line with the results.

However, the paper would need some improvements and clarifications, so I recommend its publication after minor changes.

Aspects of the manuscript that need to be revised are as follows:

-English language. Although the general level of the article is correct, it would need language revision. The manuscript has many expression and spelling mistakes that need to be revised, for example line 233 "muation" and more along the text.

-Indicate the sequence of the transit peptides.

-Figure 1B. The phylogenetic tree is very poor. Expand and introduce more species.

-In figure 3, No cyclopropane fatty acids present in E. coli were included in this figure. Please include and correct.

-Figure 5: From these data it is possible to determine the accumulated oil increment in the seeds. Please calculate and indicate. It could be an important data.

-The decrease of 22:1 could be explained from the point of view of an increase in the accumulation of C16 and C18 fatty acids caused by FAT while the synthesis of VLCFAs remains constant. Include in the discussion

Author Response

Author's Reply to the Review Report (Reviewer 1)

Comments and Suggestions for Authors

This manuscript deals with acyl-ACP thioesterases from Jatropha curcas, an oilseed of oleochemical and biotechnological interest. Thioesterases or FATs are very important enzymes in lipid synthesis and determine the flow of fatty acids exported from the plastid to the endoplasmic reticulum, where glycerolipids are synthesized.

Part by part, the introduction of the manuscript is correct as it explains well the precedents, objectives and was supported by appropriate references. The methods were correct and well described and the conclusions and discussion were in line with the results.

However, the paper would need some improvements and clarifications, so I recommend its publication after minor changes.

RESPONSE: Thanks for the referee’s positive comments that helped us a lot for improving our manuscript.

Aspects of the manuscript that need to be revised are as follows:

  1. English language. Although the general level of the article is correct, it would need language revision. The manuscript has many expression and spelling mistakes that need to be revised, for example line 233 "muation" and more along the text.

RESPONSE: Thanks for the referee’s comments. I am really sorry for the expression and spelling mistakes in the manuscript which in part because I accidentally turned off the spell checker of the Microsoft Word during the preparation of the manuscript. My co-authors and I have proofread the manuscript in detail and corrected the mistakes carefully. And a researcher in this field who had studied abroad for several years helped us with the language revision.

  1. Indicate the sequence of the transit peptides.

RESPONSE: Thanks for the referee’s constructive comments. According to the published references about FATA and FATB, we found that the sequences of the transit peptides for FATB proteins showed a higher degree of conservation, however, FATAs showed greater variations at their N-terminal sequences. And there were no sequencing evidence for the mature protein of FATA to verify the transit peptide prediction like for some FATBs, and only some computational predictions were made by using ChloroP or TargetP programme. And we also tried to predict their transit peptides using the updated version of ChloroP and TargetP, but we failed to get the results. So we did a comprehensive search from published papers for FATA and FATB, and marked the positions of the possible transit peptides for JcFATA and JcFATB in Figure S2 predicted by the previous studies [34, 35, 41, 42 ].

  1. Figure 1B. The phylogenetic tree is very poor. Expand and introduce more species.

RESPONSE: Thanks for the referee’s constructive comments. A new phylogenetic tree was generated using more FATA and FATB proteins from different plant species (Fig. 1B). And we revised the corresponding content of the text in the Materials and Methods section and the Results section accordingly in the revised manuscript.

  1. In figure 3, No cyclopropane fatty acids present in E. coli were included in this figure. Please include and correct.

RESPONSE: Thanks for the referee’s constructive comments. Cylopropane fatty acids (CFAs) occur in the phospholipids of many species of bacteria including E. coli. However, because we mainly focused on the content of saturated fatty acids and unsaturated fatty acids, we did not pay attention to the cyclopropane fatty acids when conducting this project. According to the reviewer’s comments, we rechecked our GC-MS results and did not find any these kinds of fatty acids by checking the library/ID search report for the peaks and also by checking the CAS numbers for the main CFAs in bacteria such as 19625-10-6 for cis-11,12-methyleneoctadecanoic acid (lactobacillic acid) and 4675-61-0 for cis-9,10- methyleneoctadecanoic acid. Then we checked more papers about CFAs and found the main reason for the undetection of CFAs in our results was possibly because we used acid-catalyzed methylation method (HCl-methanol solution) which could lead to the opening of cyclopropane rings to give various branched and methoxy products, and we did find the some methoxy in the GC-MS results although some of them with relatively low qual such as 64% (Grogan and Cronan, 1997. Cyclopropane ring formation in membrane lipids of bacteria. Microbiol Mol Biol Rev, 61 (4):429-441. doi: 10.1128/mmbr.61.4.429-441.). And we also compared our method with other papers with the detection of CFAs (Yu et al., 2016. Xanthomonas campestris FabH is required for branched-chain fatty acid and DSF-family quorum sensing signal biosynthesis. Sci Rep, 6: 32811. doi: 10.1038/srep32811) and found that they used the NaOH/methanol solution for fatty acid treatment. We may try measure the CFAs in our future study considering they are important for the responses to environmental stresses of bacteria as a modification of bacterial membrane lipid bilayers.

  1. Figure 5: From these data it is possible to determine the accumulated oil increment in the seeds. Please calculate and indicate. It could be an important data.

RESPONSE: Thanks for the referee’s constructive comments. We calculated the proportions of each type of fatty acids presented in Figure 5 in the total fatty acids according to the peak areas of the GC-MS results, and presented the results in Table S5 and Table S6, revised the corresponding content of the manuscript. And the results showed that although the contents for most of the fatty acids shown in Figure 5 were increased significantly, their proportion was almost no significant changes compared with the wild type. Therefore, we speculate that the total seed oil possibly increased in the ectopic expression lines of JcFATA and JcFATB.

  1. The decrease of 22:1 could be explained from the point of view of an increase in the accumulation of C16 and C18 fatty acids caused by FAT while the synthesis of VLCFAs remains constant. Include in the discussion

RESPONSE: Thanks for the referee’s constructive comments. We have added the discussion about this question in the discussion section.

Reviewer 2 Report

The authors have done extensive work on JcFATA and JcFATB expresion pattern and subcellular localization in Arabidopsis thaliana and, also JcFATA and JcFATB effect on plant growth and development as well as seed oil content were studied.

They found that JcFAT enzymes (especialy JCFATB) promote plant growth and development (plant have longer roots, stems, siliques, bigger rosette leaves and seeds compared to WT).

The Faty acid (FA) analysis of seed of the A. thaliana plants expresing JcFATA and JcFATB reveald significant stimulative effect of FAT enzymes on the acumulation of fatty acids. JcFATA showed a stronger promoting efect on accumulation of unsaturated FA, while JcFATB showed a stronger promoting effect on acumulation of saturated FA.

I have the following comments for the authors to consider:

  1. Why the authors have not checked the fatty acid composition in other tissues of Arabidopsis? It is known that FAT enzymes can influence the fatty acid composition of glycerolipids. Therefore, the FA composition should be checked at least in the leaves as well as fatty acid composition of leaf glycerolipids of wild-type, fata-oe and fatb-oe Arabidopsis plants should be determined.
  2. There are no photos confirming the phenotypic effects described in the roots, rosettes or plant habit.
  3. Due to the different possible enzymatic pathways, there is no conclusive evidence that the observed changes in plant phenotype are caused by FATA or FATB overexpressions. It is possible that increased gene expression does not correlate with increased enzymatic activity in vivo. Therefore, I recommend carrying out in vitro experiments using microsomal fractions isolated from at least seeds (although it is good to check the FAT activity in all tissues).
  4. In Section 4.9 authors said that „recombinant vectors (……….) were transformed into thumefaciens strain (……….) and then introduced into A. thaliana Col-0 as described above” The mentioned methods (transformation of A. thumefaciens and A. thaliana) are not described.
  5. What is the light intensity of plants growing on solidified agar media (in vitro) and plants growing in soil (in vivo)?

Some minor corrections:

Line 72 should be „terminated”

Line 81 should be „distinct”

Line 198 and 199 should be „hypocotyl”

Abbreviations should be expanded when they are first mentioned in the text eg DAT (line 271)

Line 485 „All primers used in this study were listed in Table S1” This should be in other section for example 4.4.
